# Effects of Resistance Training Intensity on Heart Rate Variability at Rest and in Response to Orthostasis in Middle-Aged and Older Adults

**DOI:** 10.3390/ijerph191710579

**Published:** 2022-08-25

**Authors:** Linda Li-Chuan Lin, Yi-Ju Chen, Tai-You Lin, Ting-Chun Weng

**Affiliations:** 1Institute of Physical Education, Health and Leisure Studies, National Cheng Kung University, No. 1, Ta-Hsueh Road, Tainan 701, Taiwan; 2National Sports Training Center, No. 399, Shiyun Blvd., Zuoying Dist., Kaohsiung City 813, Taiwan

**Keywords:** strength training, intensity-response, autonomic nervous system, orthostatic intolerance

## Abstract

Objective: Aging and deficits related to decreased physical activity can lead to higher risks of autonomic nervous system (ANS) dysfunction. The aim of this study was to evaluate the effects of 24 weeks of resistance training (RT) at various intensities on hemodynamics as well as heart rate variability (HRV) at rest and in response to orthostatic tests in middle-aged and older adults. Methods: Forty adults were randomized into three groups: high-intensity (HEX) (80% 1-RM) (11 female, 4 male; 60 ± 4 years); low–moderate-intensity (LEX) (50% 1-RM) (nine female, four male; 61 ± 5 years); and a control group (CON) (eight female, four male; 60 ± 4 years). The RT program consisted of nine exercises, with two sets performed of each exercise two times per week for 24 weeks. Data collected included 1-RM, heart rate, and blood pressure and HRV at rest and in response to orthostasis. Results: Both the HEX (42–94%) and LEX (31.3–51.7%) groups showed increases in 1-RM (*p* < 0.01). The HEX group showed decreases in resting heart rate (−4.0%), diastolic blood pressure (−3.2 mmHg (−4.2%)), and low frequency/high frequency (LF/HF) (Ln ratio) (*p* < 0.05). Post-study, the HEX group had higher HF (Ln ms^2^) than the CON, adjusted for pre-study value and age (*p* < 0.05). Post-study, the supine–standing ratio (SSR) of LFn (normalized unit) in the HEX group was greater than that in the LEX and CON groups, while the SSR of LF/HF in the HEX group was greater than the CON (*p* < 0.05). In conclusion, high-intensity RT can improve resting heart rate and HRV by enhancing cardiac vagal control. High-intensity RT might also improve the orthostatic response in terms of HRV. High intensity RT might assist ANS modification and could perhaps decrease the risks of cardiovascular disease and orthostatic intolerance.

## 1. Introduction

The aging process causes alterations in the cardiac autonomic nervous system. As individuals age, tonic vagal control of the heart decreases and baroreflex control of heart rate blunts at rest and while standing [1,2]. In addition, the physical changes associated with normal aging, including reduced tonic vagal control of the heart, reduced baroreflex control of heart rate while standing, and decreased mass and quality of skeletal muscle, are predisposing factors for orthostatic intolerance [3,4]. The arterial baroreflex preserves circulatory homeostasis by coordinating compensatory changes in both blood pressure and heart rate. These changes are controlled by the autonomic nervous system, which modulates parasympathetic tone and sympathetic activity [5]. Heart rate variability (HRV) measures beat-to-beat fluctuations and provides a non-invasive method of evaluating the autonomic regulation of heart rate. According to a recent meta-analysis, aerobic and/or endurance exercise training might significantly contribute to increases in RR interval and high-frequency (HF) power. Furthermore, RR intervals in older subjects showed small training responses [6].

A more recent study using supine standing ratio (SSR) and lower body negative pressure (LBNP), which shows relative changes in HRV parameters, suggested that a generally lower degree of autonomic activity combined with an attenuated parasympathetic responsiveness to postural change might somewhat explain lower orthostatic tolerance in the elderly [1]. Shi et al. reported that age-related cardiac vagal dysfunction plays a crucial role in the blunt response of arterial blood pressure regulation during hypovolemic stress. Their study concluded that a diminished heart rate response caused the elderly to experience orthostatic hypotension at the onset of orthostatic challenge [7].

A small number of studies has demonstrated that resistance training induces adaptations in cardiovascular autonomic modulation [8,9]; however, data for healthy individuals, especially the for the aged and the short-term effects of training, were limited [10,11]. Figueroa et al. studied women with fibromyalgia who had initial autonomic dysfunction and reported improved total power and cardiac parasympathetic tone after sixteen weeks of resistance training [8]. Panton et al. examined elderly subjects and showed that heart rate responses to LBNP were not altered by 12 weeks of resistance training despite increases in muscle strength and size [12]. Two meta-analyses of resistance training and blood pressure have indicated notable results after 6–30 weeks [13,14]. The potential for long-term resistance exercise training (24 weeks/almost half a year) to attenuate the decline of cardiac vagal tone in an older population needs to be revealed and verify. Resistance training has demonstrated improvements to orthostatic hypotension in the elderly through several putative mechanisms, such as enhanced baroreflex sensitivity (BRS) and sympathetic activity [15].

Perturbations and imbalances in the autonomic nervous system, consisting of increased sympathetic or reduced vagal activity, have been connected with greater risks of developing hypertension [16], cardiovascular diseases [17], and greater cardiovascular mortality [18,19]. Several studies have indicated that orthostatic hypotension is associated with higher all-cause mortality rates in older [20] and middle-aged adults [21]. Therefore, the effects of resistance training on the modulation of sympathovagal balance during orthostatic stress in middle aged and older adults must be assessed. No previous reports on this topic have assessed changes in HRV; therefore, this study provides a preliminary investigation of this issue. Training load is the essential consideration in resistance training. Despite resistance training demonstrating modifications to some cardiovascular disease risk factors, dose response relationships have not been well understood, and findings in the literature remain inconclusive [22,23,24]. Except for one study that suggested the load/volume associated with completion of 8–10 repetitions at 70% of 1RM load may provide the best stimulus for the response of postexercise hypotension [25].

The purpose of this study was to evaluate the effects of 24 weeks of resistance exercise training of various intensities on muscular strength, hemodynamics, and HRV at rest and in response to orthostatic maneuvers in middle-aged and older adults. This study postulated that resistance training improves vagal modulation, hemodynamics, and orthostatic tolerance in middle-aged and older adults. It was hypothesized that high intensity resistance training would be more effective than low-moderate intensity resistance training in modifying the specific cardiovascular disease risk factors in middle-aged and older adults.

## 2. Materials and Methods

### 2.1. Participants

Sixty non-smoking middle-aged and older adults (aged 52–68 years) (40 females, 20 males) who were recruited from community centers in southern Taiwan volunteered to participate in this study upon initial presentation. Females were eligible if they had been in menopause for more than twelve months and had not received hormone replacement therapy. The subjects’ medical histories were obtained, and a medical doctor conducted a physical examination to ensure that subjects were free from any depression, adiposity, and cardiovascular or orthopedic/neuromuscular diseases that might have hindered them from doing resistance training. Subjects with well-controlled hypertension not requiring beta-blockers, as well as those with uncomplicated diabetes (Hemoglobin A1c < 7.5%) were also included in the study. Seven subjects were taking antihypertensive medications: six in the resistance training group and one in the control group. The categories of antihypertensive medication included calcium channel antagonists [26], ACE inhibitors aldosterone receptor II blockers, and diuretics. All antihypertensive medication dosages remained unchanged during the study. Six subjects were excluded because of chronic inflammatory diseases (coronary artery disease, asthma, rheumatoid arthritis, poorly controlled diabetes, and chronic renal disease), and one female was excluded as she was already performing resistance training at the time. Subjects had not engaged in any form of resistance training during the previous year; however, they had participated in leisure activities (e.g., slow/brisk walking, qigong, tai chi, folk dancing, yoga, swimming, and hiking). All subjects were asked to maintain the same level of physical activity throughout the study. Detailed information about the study content and procedures was provided to each individual, including not ingesting foods or drinks that contain caffeine, avoiding physical activity, the benefits, inherent risks, and expected time commitments. All participants signed an informed consent document with a thorough understanding of the proposed study. The Human Experiment and Ethics Committee of National Cheng Kung University Hospital (No. ER-98-277) approved the study protocol.

### 2.2. Study Design, Group Assignments and Procedures

After baseline assessment, subjects were randomly assigned into three groups: high-intensity and low–medium-intensity resistance exercise (HEX, LEX) and the control group (CON) according to their composite strength of 1-RM of chest press and leg press. The training protocol was nine exercises—two sets two times per week for 24 weeks—while the CON maintained their regular lifestyles without doing RT. Before and after 24-week exercise intervention, anthropometric measurements (height and body mass), and strength assessment 1-RM were collected. For each subject, the measurements described in the following testing sections were carried out in the 0th and the 25th weeks. All groups were instructed not to change their diet habits.

#### 2.2.1. HRV and BP Measurements

Measurements were taken for each subject at baseline and after 24 weeks of resistance training intervention or a control period. Measurements included resting and standing HRV and blood pressure (BP) and one-repetition maximum (1-RM) testing (chest press, biceps curl, and leg press). Subjects were asked to refrain from caffeine and alcohol ingestion for 24 h prior to testing. Hypertensive subjects were asked not to take their medication during the morning prior to HRV examination. Post-training testing for all subjects took place 48 h after their last exercise training session or vigorous physical activity to avoid the acute effects of exercise. Body weight and height were measured to the nearest 0.1 kg and 0.5 cm, respectively. Body mass index (BMI) was calculated as body weight (kg) divided by the square of height (m).

#### 2.2.2. Strength Testing

Submaximal repetitions-to-fatigue tests predicted subject 1-RM using seated chest press, leg press, and biceps curl resistance exercise machines supplied by SportsArt Fitness^®^. After receiving a five-minute warm-up period, participants began the test by lifting a light load. Incremental increases in load were made according to the difficulty with which the participant performed the previous lift. The weight was increased to a load that the subject could lift one to ten times through a full range of motion with proper form. The 1-RM was then predicted using the equation developed by Brzyski: 1-RM = (weight lifted)/[1.0278 − (repetitions × 0.0278)] [26]. Prediction of 1-RM using the equation allows repeated 1-RM testing without the limitations associated with genuine 1-RM tests, which can cause fatigue and increase the risk of injury in novice weight-trainers. All measurements were recorded within three attempts. All subjects repeated 1-RM testing following 24 weeks of resistance training or a control period.

#### 2.2.3. Whole-Body Resistance Training Program

The resistance training group trained two times a week for 24 weeks, exercising on nine resistance machines, including chest press, seated back row, biceps curl, triceps extension, knee extension, back extension, abdominal curl, leg press, and leg curls. Subjects performed one set of 15 repetitions for each exercise to fatigue for the first four weeks of training and two sets of 8 (HEX group) or 13 (LEX group) repetitions (with 2–3 min of rest between sets) thereafter at the proper resistance load. The LEX and HEX groups trained at 50% and 80% of 1-RM, respectively. Total training volume for both groups was similar. To satisfy the overload rule, training load was increased when subjects in the LEX and HEX groups could perform 15 or 10 repetitions, respectively, easily and with proper form for three consecutive training days. The load was increased by 1.5–10 kg based on the upper or lower body nature of the exercise and the subject’s condition. Training logs were checked weekly, and any necessary adjustments in resistance load were made. To ensure that subjects performed two training sessions per week using the session-RPE method for training load monitoring, any missed sessions were rearranged for another day that same week. Participants had to attend > 85% of the possible training sessions to be considered compliant and qualify for the study. The duration of each exercise session was approximately 40–50 min. Subjects performed 5–10 min of warm-up and cool-down exercises before and after each session. Three physical fitness instructors supervised each entire training session.

#### 2.2.4. Autonomic Measurements

Subjects rested supine for a fifteen-minute period in a quiet, air-conditioned room, then underwent five minutes of continuous, beat-to-beat heart rate monitoring using a two-channel ECG recording one person per day in the morning (CheckMyHeart Handheld HRV, DailyCare BioMedical, Inc., Chungli, Taiwan). A digital sphygmomanometer (Model UA-787, A&D Co., Tokyo, Japan) recorded supine blood pressure after five minutes in a supine position. Subjects were then asked to move from supine to a standing position, and, once they had maintained a steady standing posture, another five-minute ECG recording was collected. Standing blood pressure was recorded at the end of a five-minute orthostatic test. Subjects were asked to avoid talking and to breathe normally during data recording and acquisition. A supine-standing-ratio (SSR), which illustrates relative changes from supine to standing posture, was calculated to express the responsiveness of some parameters of autonomic tone to orthostatic challenge [3]. Commercially available software sourced from the same company as the HRV machine performed subsequent off-line ECG signal analysis. Data was analyzed following automatic and manual exclusion of artifact and ectopic beats based on recommendations of the Task Force of the European Society of Cardiology and the North American Society of Pacing and Electrophysiology [10].

### 2.3. HRV in the Time Domain

Time domain analysis measured changes in heart rate over time, or the intervals between consecutive normal cardiac cycles. The options for manual intervention by the researcher form key components for confident HRV analysis. If an RR interval differs from the local average of more than 0.25 s threshold value, the interval is identified as an artifact and is marked for correction. RMSSD (square root of the mean of the sum of squared differences between adjacent normal-to-normal intervals) (ms) is one of the most common parameters for interval differences, corresponding to short-term HRV changes. RMSSD reflects alterations in autonomic tone that are predominantly vagally mediated.

### 2.4. HRV in the Frequency Domain

Power spectral density analysis assessed the frequency domain. The data was processed rapidly using the Fourier transformation, which is characterized by discrete peaks for the several frequency components. The total power (TP) of RR interval variability is the total variance. High-frequency (HF) oscillations of HRV (0.15–0.40 Hz) are generally defined as markers of vagal modulation. The low-frequency (LF) (0.04–0.15 Hz) power is modulated by both the sympathetic and parasympathetic nervous systems. HF and LF powers were expressed in absolute values (ms^2^) and in normalized units (nu) to reduce the effects of noise due to artifacts and to minimize the effects of changes in total power on the HF and LF components. The LF/HF ratio reflects the global sympathovagal balance and serves as a measure of this balance.

## 3. Statistical Analyses

Natural log transformations (Ln) were used for data that were not normally distributed. All data were reported as mean ± SD or SEM. One-way ANOVA detected differences between groups at baseline and percentage changes in muscular strength after training. Student paired *t*-tests determined whether within-group changes occurred. A two-way analysis of covariance (ANCOVA) was performed on outcome variables post-study, with the pre-study value and age as the covariates. A Scheffé post hoc test analyzed the predicted means generated when the ANCOVA revealed that the covariate significantly contributed to the outcome. Bonferonni’s correction adjusted for multiple comparisons. SPSS software version 15.0 for Windows (SPSS, Chicago, IL, USA) performed statistical analyses. Significance was set at alpha < 0.05.

## 4. Results

### 4.1. Subjects

Figure 1 shows a flowchart of the present study. Thirty-four training group subjects and thirteen control subjects completed the study. Two training group subjects and three CON subjects dropped out during training. Six training group subjects and one CON subject were not included in final data analyses. Therefore, all analyses were completed on 12 CON (eight female, four male), 13 LEX (nine female, four male), and 15 HEX (11 female, 4 male) participants. The study population size in this intervention study considered a sample size calculation. This was based on demonstrating the effect size with 95% power at a significance level of 5% for 3 groups, resulting in a size of at least 35 subjects (calculated using G*Power 3. 0, GPowerNT.exe). Compliance was >90% in all remaining training subjects. No adverse events related to resistance exercise were noted during the study. There were no significant differences between groups for age, body weight, body height, or body mass index at initial presentation (*p* ≥ 0.05) (Table 1).

### 4.2. Effects of Resistance Training on Muscle Strength

Table 2 presents changes in upper body strength (chest press, biceps curl) and lower body strength (leg press) from pre- to post-study. No significant differences between groups were observed at baseline. Upper body strength (*p* < 0.001) and lower body strength (*p* < 0.01) significantly improved with training for both the LEX and HEX groups. By contrast, the control group showed decreases in all three strength variables, especially in leg press strength (*p* < 0.01). Both the LEX and HEX groups showed significantly greater percentage change (%) in the three tests than the CON group; however, there was no significant difference between the LEX and HEX groups.

### 4.3. Changes in Hemodynamic and HRV Measures

Table 3 shows resting supine HRV data, and Table 4 shows hemodynamic data. There were no significant differences between groups in any time or frequency domain measures of HRV, heart rate, or blood pressure (BP) at baseline. There was a significant decrease from pre- to post-study in LH/HF (Ln ratio) in the HEX group (*p* < 0.05). In the time domain variables of HRV, the LEX and HEX groups showed a non-significant increase in RR interval (ms^2^) and RMSSD (Ln ms), while the CON group showed a slight decrease. Post-study, only the HEX group had a significantly higher HF (Ln ms^2^) compared to the CON group (*p* < 0.05), adjusted for pre-study level and age. Resting heart rate significantly decreased after training in the HEX group only (*p* < 0.05). Post-study, only the HEX group had a significantly lower heart rate compared with the LEX group (*p* < 0.05), adjusted for pre-study level and age. Both training groups showed a non-significant decrease in systolic BP from pre- to post-study, with decreases of −2.8 mmHg (−2.1%) and −4.3 mmHg (−3.3%) for the LEX and HEX groups, respectively (*p* > 0.05). Although the LEX group had a decrease in supine diastolic BP (−3.9 mmHg (−4.6%)) after training, the change in supine diastolic BP was significant only for the HEX group (−3.2 mmHg (−4.2%)) (*p* < 0.05).

### 4.4. Autonomic Reflex Response to Orthostatic Maneuver

Figure 2 presents the results for SSR, which show relative changes in HRV, BP, and HR parameters from a supine to a standing position. No significant differences in the SSR of LFn, HFn (in normalized unit), LF/HF, HR, SBP, or DBP were noted between the three groups at baseline. No significant pre- to post-study changes in the SSR of LFn, HFn, or LF/HF were found for any group. At post-study, the HEX group had significantly higher SSR of LFn, and LF/HF levels (%) compared with the CON and LEX groups, adjusted for pre-training level and age (*p* < 0.05).

No significant pre- to post-study changes in the SSR of HR, SBP, or DBP were found for any of the groups. Post-study, no significant differences were noted between the three groups.

## 5. Discussion

The main finding of this study was that 24 weeks of resistance training resulted in increased upper and lower body strength in both LEX and HEX groups agree with previous studies [27]. The HEX group also showed improvements in resting heart rate, diastolic blood pressure, and specific HRV measures compared to the CON and LEX groups. The novel finding of this study was that only high-intensity resistance training had positive effects on the autonomic responses to orthostatic maneuvers. Therefore, high-intensity resistance training may have a positive effect on cardiac autonomic modulation at rest and under orthostasis for middle-aged and older adults.

### 5.1. Heart Rate and Blood Pressure at Rest and in Response to Orthostatic Maneuvers

Twenty-four weeks of resistance exercise training resulted in reduced resting heart rate in the HEX group. The evidence specifically concerning the effect of resistance training on resting heart rate is currently limited. Two meta-analyses investigating the effects of resistance training on blood pressure reported that moderate intensity resistance training might assist blood pressure control, although they did not find a significant change in resting heart rate [13,14]. Vincent et al. found that significant improvements in aerobic capacity and treadmill exercise to exhaustion time can be obtained in older adults as a result of either high- or low-intensity resistance exercise, with a non-significant decrease in heart rate noted only in the high-intensity group [28]. The present study prescribed exercise intensity similar to Vincent’s protocol, although subjects in the present study performed two sets of each exercise two times per week, while subjects in Vincent’s study performed one set three times per week. The total amount of load was greater in the present study; therefore, training volume might be one reason for the significant decrease in resting heart rate.

Although a maximal aerobic test was not used to assess cardiorespiratory adaptations, it might be possible to consider improved cardiorespiratory capacity as a likely contributor to resting bradycardia. In addition, the relative improvements in vagal tone reflected in HRV variables in the HEX group might be another explanation for the resting bradycardia. The orthostatic responses of heart rate were significant both pre- and post-study in all three groups; the changes in heart rate responses of subjects in the LEX and HEX groups showed that cardio acceleration was not increased after resistance training. Moving from supine to an upright posture is accompanied by a significant amount of blood movement from the chest cavity into the lower body, which can result in a reduction in stroke volume and a decrease in arterial pressure [29,30]. When normal individuals assume an upright position, systolic BP drops by no more than 5–10 mmHg, diastolic BP rises and pulse rate increases by 10–25 beats per minute. In the present study, baseline heart rate responses to the orthostatic challenge were not abnormal, which showed no difference between the three groups of subjects.

The high-intensity training group showed significant improvements in resting diastolic BP. The mean resting diastolic BP decreased by 3.9 mmHg (4.6%) and 3.2 mmHg (4.2%), while mean resting systolic BP decreased by 2.8 mmHg (2.1%) and 4.3 mmHg (3.3%), for the LEX and HEX groups, respectively. The effects of resistance training on blood pressure in the present study are consistent with the results of previous research. Two meta-analyses of the role of resistance training in controlling blood pressure have shown that the range for the reduction of resting systolic BP and diastolic BP is 3–3.5 mmHg (2% and 4%, respectively) [13,14]. One commonly suggested mechanism is a reduced sympathetic tone after resistance training. Anton et al. reported that resistance training increased blood flow and vascular conductance in limbs of older adults [1]. It is also well known that small reductions (3 mmHg) in the average blood pressure of a population decrease the incidence of coronary heart disease and stroke [31], suggesting that the similar improvement in blood pressure after resistance training found in the present study might be beneficial for older subjects.

### 5.2. Vagal Tone Modulation at Rest

Cardiac vagal modulation of heart rate appears to be enhanced under resting conditions after a period of exercise training. Several studies have shown that resting HRV is greater in exercise-trained subjects than those who are sedentary [32,33,34]; however, there is some evidence that does not support the acute and chronic effects of resistance exercise on HRV in older adults [27]. This lack of consistent findings is probably for at least two reasons: inter-study differences in the stimulus of the exercise training (i.e., differences in the duration, intensity, frequency, and length of exercise training) and the law of initial value [32]. Therefore, a significant enhancement of autonomic modulation of heart rate would be likely to occur under the following conditions: recruited subjects have very low baseline levels and undergo sufficiently intense and lengthy exercise training and highly exercise-trained subjects are compared with very sedentary adults. The amount of information in the current literature about the effects of resistance training on the autonomic nervous system is limited. In addition, most of the existing studies that show positive effects of resistance training on HRV included subjects with low baseline HRV levels, such as women with fibromyalgia [8] and patients with chronic heart failure [9]. Studies that enrolled healthy subjects [35,36] or that lasted for a short duration (four to six weeks) [37] have shown either no significant effects or even negative effects of resistance training on HRV.

To the best of the authors of this study’s knowledge, no studies have focused on the effects of long-term (24 weeks) resistance training of various intensities on HRV, especially in an older population. The present study showed that LF/HF (Ln ratio) in the HEX group significantly decreased from pre- to post-study. In addition, the HEX group had significantly higher HF power (Ln, ms^2^) at post-study compared with the CON group. The LEX group showed no significant modifications in the measured HRV variables after resistance training. The main reason for this could be that the proposed low–moderate-intensity resistance training was not intense enough to induce any modifications in the autonomic control of the heart, although the intensity satisfied the guidelines for exercise prescription in older subjects [38]. Future research may therefore consider longer training durations to observe if any significant results can be obtained. Current evidence in the literature shows that the positive effects of exercise on HRV are generally obtained from at least moderate-intensity [39,40] and long-duration endurance exercise [41,42]. Therefore, only subjects in the high-intensity resistance training group showed significant changes in specific HRV measures compared with the CON group. Chronic endurance exercise training elicits improvements in HRV thought to be partly due to an augmented cardiac vagal modulation, such as significant increases in RR interval and HF power [6]. Accordingly, high-intensity resistance training in the present study might improve cardiac parasympathetic tone in older adults, which is a predictor of cardiac morbidity and mortality. Therefore, resistance training does not appear to influence sympathoadrenal system activity and cardiac autonomic status at rest [3].

### 5.3. HRV Responses to the Orthostatic Challenge

The analyses were performed from linear time domain models—rMSSD (the square root of the mean squared differences of successive R–R intervals)—and in the frequency domain through spectral analysis, low frequency components (LF) being representative of the sympathetic component of the system, high frequency (HF) being representative of the parasympathetic component of the system, and the ratio LF/HF representing the sympathovagal balance. Results for HRV variables reflected the cardiac autonomic response to postural change. In the LEX and HEX groups, both pre- and post-study LFn (nu) and LF/HF ratio (Ln) were significantly higher than the CON group from supine to a standing position, while RMSSD (Ln, ms) and HFn (nu) decreased significantly under orthostasis (*p* < 0.01). The results substantiate that high-intensity resistance training can improve the autonomic modulation in which orthostasis is associated with vagal inhibition and sympathetic enhancement [5].

A trend towards increasing SSR of LFn and decreasing SSR of HFn appeared in the LEX and HEX groups after the 24-week study period; however, post-study results showed that only the SSR of LFn and LF/HF for the HEX group was significantly improved compared with the LEX and CON groups (*p* < 0.05). During an orthostatic challenge, hypovolemic stress occurs, in which the movement of blood to peripheral veins is suggested to occur via gravity. This predominantly stimulates the carotid baroreceptors, which can influence heart rate, peripheral resistance, and stroke volume. It has been proposed that resistance training might improve orthostatic tolerance by improving baroreflex sensitivity (BRS) [8]. In particular, HEX is a form of intense exercise that is prescribed in various modalities and durations with an increase in sympathetic modulation and a decrease in parasympathetic modulation after HEX. This finding of LFn (nu) and LF/HF ratio (Ln) increase may have an important clinical meaning; to increase in sympathetic activation combined with a reduction of parasympathetic activation may increase the risk of cardiovascular events in older adults.

In response to the change in blood pressure, afferent nerves in the carotid region send signals to the brainstem that act centrally to modulate sympathovagal outflow to the heart, such as by reduction of cardiac vagal tone. In addition, the sympathetic noradrenergic system innervating the heart and peripheral blood vessels activates under the stimulus of orthostasis [29]. Many of these compensatory changes are coordinated through the activities of the autonomic nervous system, such as the modulation of sympathetic parasympathetic balance. It has been proposed that resistance training might improve orthostatic tolerance through the suggested mechanisms, such as by increasing strength while decreasing peripheral resistance and preserving arterial pressures to improving baroreflex sensitivity (BRS) [11].

The majority of previous studies reporting the effects of resistance training on orthostatic tolerance assessed the baroreceptor cardiac reflex response using the LBNP or head up tilt (HUT) test; however, the findings of these studies were conflicting. Tatro et al. reported that nineteen weeks of high resistance, lower extremity exercise training had no significant effect on LBNP tolerance [43]. Carroll et al. found that six months of endurance training in elderly subjects, alone or in combination with selected resistance exercises, had no significant effects on blood pressure controlling mechanisms measured using HUT [44]. McCarthy et al. reported that twelve weeks of resistance training did not alter BRS of heart rate and maximal LBNP tolerated in young adults [45], while Panton et al. showed that twelve weeks of resistance training did not alter heart rate responses to LBNP in the elderly [12]. Conversely, Lightfoot et al. indicated that twelve weeks of resistance training increased LBNP tolerance, with a group of chronically resistance-trained individuals showing an even greater increase in LBNP tolerance [46]. The HUT test and LBNP do not completely represent the physiological process of postural change from a supine to a standing position, especially given that other physiological compensations—such as the local axon reflex and the myogenic response—are not considered due to lost activation of muscles in the lower limb. The present study is the first to use HRV parameters to present the effects of resistance training on orthostatic responses.

Aging-related changes in cardiac autonomic function comprise a host of changes in the autonomic nervous system, such as diversity of the baroreceptor response, peripheral and central neural integration, and sinoatrial node reaction. At least two possible mechanisms are responsible for the exercise-induced increase in BRS. One possible mechanism is changes in vessel wall compliance of the carotid artery, where pressure-sensitive baroreceptors are located. However, the impacts of resistance training on central arterial compliance are in dispute of a number of studies, with some earlier reports finding either no change or improvements in arterial compliance and vascular function [47,48,49], while others suggested that prolonged resistance training might be associated with decreases in arterial compliance [50,51].

The other possible mechanism is neural changes in the baroreflex arc. Studies have proposed that both vascular and neural deficits contribute to age-related declines in cardiovagal baroreflex gain. These studies found that neural BRS, estimated by carotid arterial diameter, and RR intervals in trained elderly men were higher than in sedentary peers, suggesting that long-term physical activity attenuates this decline by maintaining neural vagal control [52]. Analysis of HRV is a simple, non-invasive method to evaluate the sympathovagal balance at the sinoatrial level and may be a surrogate for these complex vascular–neural interactions [3]. The present study found a significant reduction in blood pressure after resistance training; therefore, some improvement in vascular health might have occurred. Based on this result, resistance training is not likely to decrease the compliance of the carotid artery. In addition, a significant enhancement of resting vagal tone was noted in the HEX group after training. Therefore, the trend toward increasing SSR of LFn and decreasing SSR of HFn in both resistance training groups, as well as the significantly higher post-study SSR of LF/HF for the HEX group compared with the CON group, might be more likely to have resulted from improved neural vagal modulation of the baroreflex arc. The blood pressure results of this study are partially in agreement with previous studies [47,49]. These studies showed greater and longer post training hypotension responses with higher load intensities, which can play a key role in chronic blood pressure and cardiovascular risk reduction [47]. Furthermore, half a year should be able to achieve the stabilizing effects of the intervention since resistance training also changes weight, and lean mass, thus aiding the metabolism and mitigating the risk factors of cardiovascular disease.

### 5.4. Limitations

There are several limitations in the present study, which provide some promising avenues for future research. The possibility of a small sample size-related error might have limited the results. HRV changes are associated with cardiorespiratory fitness; therefore, this study did not assess subjects’ fitness levels using a maximal aerobic test in order to control bias in the experimental design. In addition, this study assessed muscle strength using the 1-RM test to provide the maximum load with which an individual could perform a complete repetition of a given exercise instead of isokinetic dynamometer. Isokinetic testing was considered as it provides a general ability to produce force in a given joint, but that is not the goal of this study [53]. The goal of this study was to determine strength changes over time in a given exercise such as the chest press or leg press, then testing should be carried out using the testing modalities (1-RM). This study enrolled healthy subjects, none of whom were diagnosed with orthostatic intolerance at baseline; therefore, these results cannot be applied as generalizations to older people with initial orthostatic intolerance.

## 6. Conclusions

In conclusion, the findings of the present study suggest that 24 weeks of resistance training results in improved upper and lower body muscular strength and moderate blood pressure control. In addition, resistance-training-induced adaptations in the autonomic nervous system might alter the neural regulatory mechanisms controlling cardiac and vascular functions. The high-intensity resistance training would be more effective than low-moderate intensity resistance training to cause a greater improvement in resting heart rate, cardiac vagal control, and orthostatic tolerance, as reflected by HRV measurements. Resistance training of sufficient intensity might therefore be a safe and beneficial lifestyle adaptation to improve muscular strength, the cardiovascular system, cardiac autonomic health, and orthostatic tolerance in middle-aged and older adults.

Further studies are required to establish the relationship and possible mechanisms between resistance training and improved autonomic modulation at rest and under orthostasis. In addition, the effects of age and baseline condition of subjects, exercise training protocol, and study design on changes in vascular–neural interactions require further investigation in order to provide more conclusive and comparable evidence.

## Figures and Tables

**Figure 1 ijerph-19-10579-f001:**
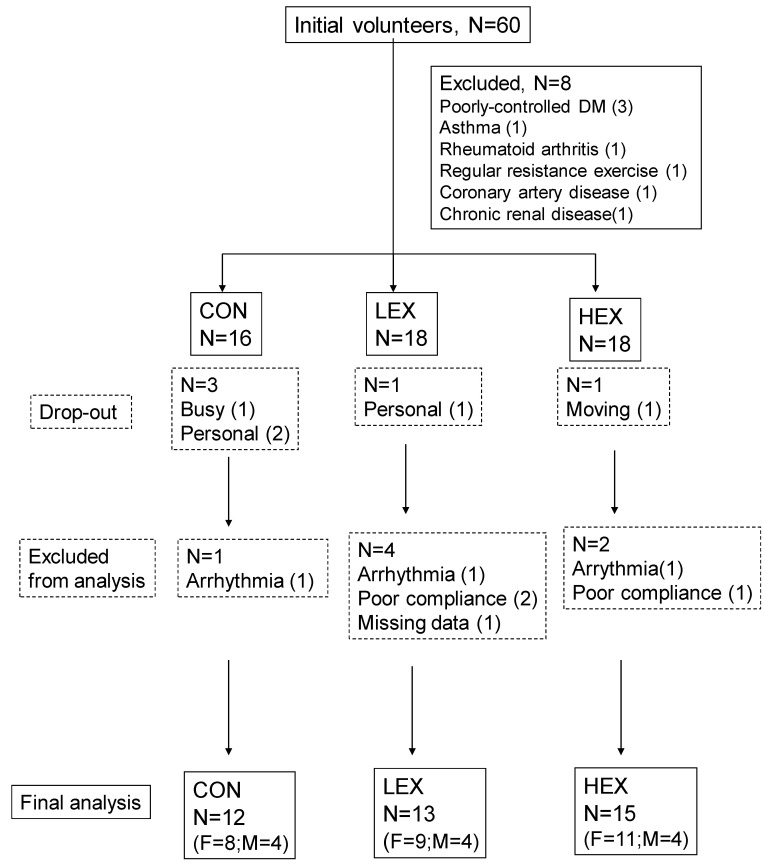
Flowchart of the study. F, female subjects; M, male subjects. CON, control group; LEX, low–middle-intensity training group; HEX, high-intensity training group.

**Figure 2 ijerph-19-10579-f002:**
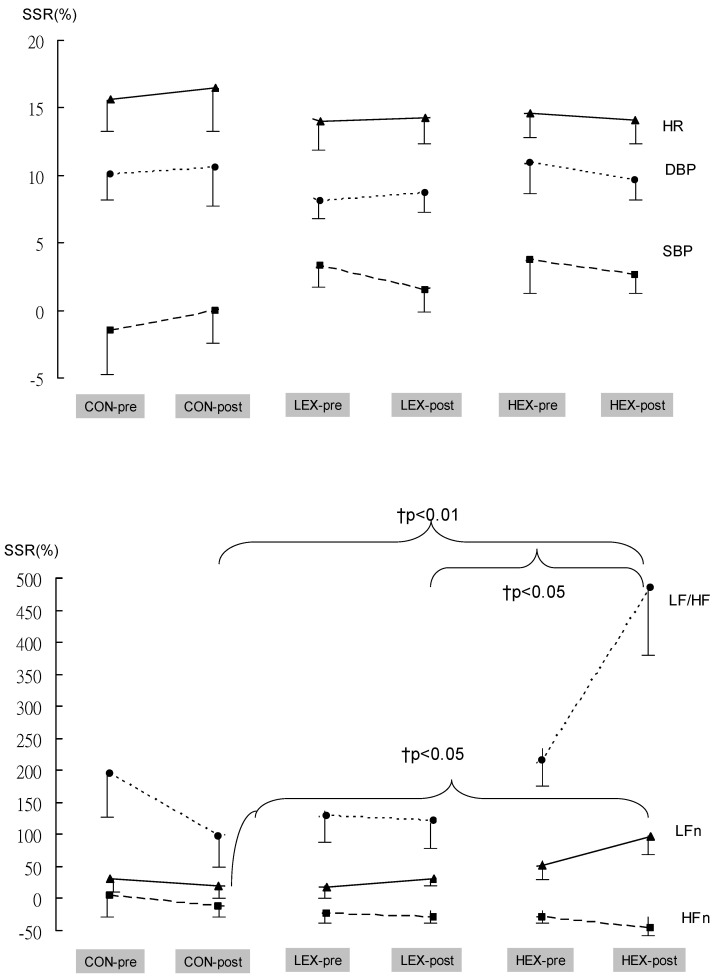
Changes in supine–standing ratio (SSR) as percentage differences between cardiovascular parameters and HRV parameters in pre- and post-study measurements for the three groups. HFn = HF power in normalized units; LFn = LF power in normalized units; HR = heart rate; SBP = systolic blood pressure; DBP = diastolic blood pressure. † *p* < 0.05, vs. LEX or CON by analysis of covariance with pre-training level and age as covariates.

**Table 1 ijerph-19-10579-t001:** Subject Characteristics.

Variable	CON (*N* = 12) (8 Female, 4 Male) (H/T 1, DM 1)	LEX (*N* = 13) (9 Female 9, 4 Male) (H/T 3, DM 1)	HEX (*N* = 15) (11 Female, Male) (H/T 3, DM 1)
Age (years)	59.8 ± 4.2	61.1 ± 5.3	59.7 ± 3.7
Body height (cm)	155.5 ± 6.4	157.8 ± 8.7	159.7 ± 7.2
Body weight (kg)	56.9 ± 13.8	58.6 ± 9.3	57.8 ± 12.4
BMI (kg/m^2^)	23.5 ± 5.1	23.4 ± 1.9	22.5 ± 3.1

Values are expressed as mean ± SD; CON = control group; LEX = low–moderate-intensity training group; HEX = high-intensity training group; H/T = hypertension; DM = diabetes mellitus.

**Table 2 ijerph-19-10579-t002:** Changes in muscular strength for the CON, LEX, and HEX groups following 24 weeks of resistance training or a control period.

Variable	1-RM	CON (*N* = 12)	LEX (*N* = 13)	HEX (*N* = 15)
Pre-training (kg)	biceps	14.2 ± 2.6	12.6 ± 1.9	9.6 ± 2.2
	chest press	33.5 ± 5.0	30.7 ± 4.6	25.4 ± 3.8
	leg press	113.6 ± 17.5	103.9 ± 10.1	102.6 ± 9.8
Post-training (kg)	biceps	12.8 ± 2.4	17.5 ± 2.8 ^§§^	14.2 ± 2.1 ^§§^
	chest press	29.7 ± 5.0	41.5 ± 4.5 ^§§^	35.8 ± 4.0 ^§§^
	leg press	93.8 ± 12.6 ^§^	126.6 ± 7.3 ^§^	134.5 ± 8.1 ^§^
Percent change (%)	biceps	−7.6 ± 4.4	43.4 ± 9.6	94.4 ± 35.5 ^¶¶^
	chest press	−11.8 ± 6.4	51.7 ± 16.6 ^¶¶^	55.2 ± 12.2 ^¶¶^
	leg press	−13.8 ± 5.2	31.3 ± 11.1 ^¶^	42.0 ± 12.7 ^¶¶^

Values are expressed as mean ± SEM. CON, control group; LEX = low–medium-intensity training group; HEX = high-intensity training group. ^§^ *p* < 0.01, ^§§^ *p* ≤ 0.001 vs. pre-training; ^¶^ *p* < 0.05, ^¶¶^ *p* ≤ 0.01 vs. CON using one-way ANOVA.

**Table 3 ijerph-19-10579-t003:** Changes in resting HRV measures for the CON, LEX, and HEX groups following 24 weeks of resistance training or a control period.

Variable	CON (*N* = 12)	LEX (*N* = 13)	HEX (*N* = 15)
RR interval (ms)			
Pre-training	956.4 ± 32.0	899.7 ± 15.9	951.4 ± 35.9
Post-training	954.5 ± 24.8	903.6 ± 18.5	1001.7 ± 39.6
RMSSD (Ln ms)			
Pre-training	3.05 ± 0.20	2.89 ± 0.14	3.24 ± 0.16
Post-training	3.00 ± 0.13	2.94 ± 0.13	3.37 ± 0.16
Total power (Ln ms^2^)			
Pre-training	6.47 ± 0.25	6.22 ± 0.20	6.44 ± 0.25
Post-training	5.94 ± 0.23	6.18 ± 0.21	6.41 ± 0.24
Low-frequency (nu, %)			
Pre-training	62.8 ± 5.8	57.5 ± 4.1	52.2 ± 5.2
Post-training	57.1 ± 5.2	54.6 ± 4.7	41.5 ± 4.2
Low-frequency (Ln ms^2^)			
Pre-training	4.92 ± 0.31	4.55 ± 0.26	4.92 ± 0.26
Post-training	4.33 ± 0.28	4.51 ± 0.23	4.75 ± 0.31
High-frequency (nu, %)			
Pre-training	37.1 ± 5.8	42.5 ± 4.1	47.8 ± 5.2
Post-training	42.9 ± 5.2	45.4 ± 4.7	58.5 ± 4.2
High-frequency (Ln ms^2^)			
Pre-training	4.27 ± 0.37	4.22 ± 0.30	4.80 ± 0.34
Post-training	3.99 ± 0.27	4.27 ± 0.28	5.12 ± 0.29 ^‡^
LH/HF (Ln ratio)			
Pre-training	0.64 ± 0.29	0.33 ± 0.18	0.12 ± 0.24
Post-training	0.34 ± 0.24	0.24 ± 0.22	-0.37 ± 0.18 ^§^

Values are expressed as mean ± SEM. CON = control group; LEX = low–medium-intensity training group; HEX = high-intensity training group; RMSSD = root mean square SD; Ln = natural logarithm; LF/HF = low-frequency/high frequency; nu = normalized units. ^§^ *p* < 0.05, vs. pre-training; ^‡^ *p* < 0.05, vs. CON by analysis of covariance with pre-training level and age as covariates.

**Table 4 ijerph-19-10579-t004:** Changes in supine hemodynamic measures for the CON, LEX, and HEX groups following 24 weeks of resistance training or a control period.

Variable	CON (*N* = 12)	LEX (*N* = 13)	HEX (*N* = 15)
Heart rate (beats min^−1^)			
Pre-training	62.4 ± 2.1	64.5 ± 1.3	63.9 ± 2.2
Post-training	63.3 ± 2.2	65.9 ± 1.8	60.9 ± 2.3 ^§,^*
Percentage change (%)	2.1 ± 3.3	2.2 ± 2.6	−4.6 ± 3.0
Systolic BP (mmHg)			
Pre-training	117.0 ± 4.5	118.6 ± 3.9	114.5 ± 4.0
Post-training	110.7 ± 4.8 ^§^	115.9 ± 3.7	110.1 ± 3.3
Percentage change (%)	−5.4 ± 1.7	−2.1 ± 1.5	−3.3 ± 1.9
Diastolic BP (mmHg)			
Pre-training	73.3 ± 3.0	76.2 ± 2.6	69.9 ± 2.5
Post-training	70.4 ± 3.0	72.3 ± 2.6	63.9 ± 2.2 ^§^
Percentage change (%)	−3.5 ± 1.5	−4.6 ± 2.8	−4.2 ± 1.4

Values are expressed as mean ± SEM; BP = blood pressure. * *p* < 0.05 vs. LEX by analysis of covariance with pre-training level and age as covariates. ^§^ *p* < 0.05 vs. pre-training.

## Data Availability

Not applicable.

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
