# Peer review of "Effects of Resistance Training Intensity on Heart Rate Variability at Rest and in Response to Orthostasis in Middle-Aged and Older Adults"

_ijerph, 2022, doi:10.3390/ijerph191710579_

Round 1

Reviewer 1 Report

Effects of resistance training intensity on HRV at rest and in response to orthostasis in middle-aged and older adults

I would like to thank the Editor for the opportunity to review this paper.

The topic is very interesting and important because of the aging society. The study is interesting, and the results are promising. However, the main shortcoming of this paper is that it is not precise enough and confusing for the reader. The authors are switching between the topics. The main research question might be if low or high intensity resistance training elicit improvements in cardiovascular health. However, too many other aspects were raised and the authors do not adequately answer the main question. As the study is interesting, I recommend major revisions before considered for publication.

General remark

The order of the references are not correct. I would recommend to substitute “subjects” with “participants” as this is more common today. The authors switch between cardiac autonomic activity, tone, control, and modulation. There should be an uniform term based on current literature. The discussion should be shortened. One page less would be still enough.

Abstract

The abstract is well-written, precise and the reader gets a good overview of the study.

Introduction

Line 41+42: there should be a reference after the sentence “It is widely believed …”.

In general, the introduction is rather extensive, and the studies are described with too many details. The authors sometimes jump between the topics. For the reader who is unfamiliar with the ANS, it is difficult to follow. I would recommend being more precise and shorten the introduction. For example, it is not necessary to describe all studies in such detail.

Material and methods

The authors should state how and from where the participants were recruited?

Were depression or adiposity an exclusion criteria?

How long was the resting time between the sets?

I miss some information concerning the autonomic measurements:

  • Time of day of the measurements
  • Sampling rate
  • In my opinion it is not sufficient to write “… automatic and manual exclusion of artifact and ectopic beats based on recommendations of the Task Force”. The criteria for artifacts and ectopic beats should be clearly stated (e.g., specific length of RR interval).
  • Artifact cleaning method?
  • Were participants with specific number of artifacts excluded?
  • Why did the authors use LF/HF as marker of sympatho-vagal balance although current literature does not support this assumption anymore (e.g., Billman, 2013, doi: 10.3389/fphys.2013.00026)?

Results

Page 8, line 298: “figure not shown” is not necessary or the authors include a figure.

Figure 2 is confusing. The authors should look for a diagram which is more intuitive for the reader.

Discussion

As already noted above, the discussion is too long. The authors should more focus on the assessed parameters in this study. E.g., VO2 max was not assessed and the sentences about this parameter could be deleted. Some sentences could also be deleted as they are only a repetition of the results (e.g., page 10, lines 356-359).

Moreover, as only relatively healthy older adults were measured, discussion should focus only on this group.

Section 5.2. should describe “vagal tone”, so why were the results of LF discussed? LF is no parameter for the vagal tone.

Section 5.3. is very long and confusing. The authors should be more precise. Some parts can be removed because they are basic physiological knowledge. Some parts are not relevant for this study.

The sense of the last paragraph (line 502-509) is not clear. Is it a summary of the study or of the literature (although no references were included)?

Line 522-533: this is the conclusion. It should be under a separate section.

As the main aim of the study was to compare low and high intensity resistance training, I am wondering why this aspect was so little discussed. It is clear that any kind of training elicits effects in untrained older adults. However, one important research question is, what kind of training (intensity, load, type) is adequate to elicit improvement in cardiovascular and autonomic health. In my opinion, the reader is interested in this question and therefore, the authors should focus on this in the discussion.

Author Response

We would like to thank the editor and reviewers for the time to review our manuscript. We replied the questions and suggestion in the following.

No.

Reviewer comment

Author response

Reviewer #1

The topic is very interesting and important because of the aging society. The study is interesting, and the results are promising. However, the main shortcoming of this paper is that it is not precise enough and confusing for the reader. The authors are switching between the topics. The main research question might be if low or high intensity resistance training elicit improvements in cardiovascular health. However, too many other aspects were raised and the authors do not adequately answer the main question. As the study is interesting, I recommend major revisions before considered for publication.

We appreciate that Reviewer #1 recognized the importance of aging and if low or high intensity resistance training elicit improvements in cardiovascular health. We will respond to all reasonable comments and the main question in the following.

1. General remark

The order of the references are not correct. I would recommend to substitute “subjects” with “participants” as this is more common today. The authors switch between cardiac autonomic activity, tone, control, and modulation. There should be an uniform term based on current literature. The discussion should be shortened. One page less would be still enough.

We appreciate the recognition from the Reviewer #1.The abstract section will not be modified for this time.

2. Introduction

Line 41+42: there should be a reference after the sentence “It is widely believed …”.

In general, the introduction is rather extensive, and the studies are described with too many details. The authors sometimes jump between the topics. For the reader who is unfamiliar with the ANS, it is difficult to follow. I would recommend being more precise and shorten the introduction. For example, it is not necessary to describe all studies in such detail.

We thank Reviewer #1 for the comment. As we responded to Comment #1, we carefully reviewed the Introduction multiple times and made significant revisions, and shorten the introduction. We had tried to precise enough and logical structure of the introduction. We hope that is satisfactory. ( Line 33-66, red words) Line 33-66

3. Material and methods

1.The authors should state how and from where the participants were recruited?

2.Were depression or adiposity an exclusion criteria?

3.How long was the resting time between the sets?

4.I miss some information concerning the autonomic measurements:

5.Time of day of the measurements

6.Sampling rate

Sampling rate

In my opinion it is not sufficient to write “… automatic and manual exclusion of artifact and ectopic beats based on recommendations of the Task Force”. The criteria for artifacts and ectopic beats should be clearly stated (e.g., specific length of RR interval).

Artifact cleaning method?

Were participants with specific number of artifacts excluded?

Why did the authors use 7.LF/HF as marker of sympatho-vagal balance although current literature does not support this assumption anymore (e.g., Billman, 2013, doi: 10.3389/fphys.2013.00026)?

We thank Reviewer #1 for the reminds. We will respond to all reasonable comments and the main question in the following.

1. Sixty non-smoking middle-aged and older adults (aged 52-68 years) (40 females, 20 males) were recruited from community. (Line 139)

2. The depression and adiposity were the exclusion criteria.(Line 143)

3. Subjects performed one set of 15 repetitions for each exercise to fatigue for the first four weeks of training, and two sets of 8 (HEX group) or 13 (LEX group) repetitions (with 2–3 min of rest between sets) thereafter at the proper resistance load. (Line 202)

4. Can be more explained and highlighted

5. in the morning, (Line 217-219)

6. We thank Reviewer #1 for pointing them out. We had made the corrections accordingly for artifacts and ectopic beats should be clearly stated (e.g., specific length of RR interval) (Line 235-238,  248-251). Seven subjects were excluded because of chronic inflammatory diseases (coronary artery disease, asthma, rheumatoid arthritis, poorly-controlled diabetes, and chronic renal disease) (Figure 1.)

7. The LF/HF ratio reflects the global sympatho-vagal balance, and serves as a measure of this balance. Whether or not this theory is still valid is still a matter of opinion.

There are also continuously related literatures to support the adoption. After consideration, we still follow the original experimental design for analysis.

Shaffer F, Ginsberg JP (2017) An overview of heart rate variability metrics and norms. Front public Heal 5:258. https://doi.org/10.3389/fpubh.2017.00258

PL Latchman, G Gates, J Pereira, R Axtell R(2020) The association between sympatho-vagal balance and central blood pressures. Physiol Int. 107(1):155-165. doi: 10.1556/2060.2020.00005

4.

Results

Page 8, line 298: “figure not shown” is not necessary or the authors include a figure.

Figure 2 is confusing. The authors should look for a diagram which is more intuitive for the reader.

1.We thank Reviewer #1 for pointing them out. We had added the title for figure1.

.

2. We thank Reviewer #1 for pointing this out.  Figure 2 can more intuitively show the Changes in the supine-standing ratio (SSR) as percentage differences between cardiovascular parameters and HRV parameters in pre-and post-study measurements for the three groups. Therefore we decided to keep the same results.

5.

Discussion

1.As already noted above, the discussion is too long. The authors should more focus on the assessed parameters in this study. E.g., VO2 max was not assessed and the sentences about this parameter could be deleted. Some sentences could also be deleted as they are only a repetition of the results (e.g., page 10, lines 356-359).

2.Moreover, as only relatively healthy older adults were measured, discussion should focus only on this group.

3.Section 5.2. should describe “vagal tone”, so why were the results of LF discussed? LF is no parameter for the vagal tone.

4.Section 5.3. is very long and confusing. The authors should be more precise. Some parts can be removed because they are basic physiological knowledge. Some parts are not relevant for this study.

5.The sense of the last paragraph (line 502-509) is not clear. Is it a summary of the study or of the literature (although no references were included)?

Line 522-533: this is the conclusion. It should be under a separate section.

6.As the main aim of the study was to compare low and high intensity resistance training, I am wondering why this aspect was so little discussed. It is clear that any kind of training elicits effects in untrained older adults. However, one important research question is, what kind of training (intensity, load, type) is adequate to elicit improvement in cardiovascular and autonomic health. In my opinion, the reader is interested in this question and therefore, the authors should focus on this in the discussion.

1.We thank Reviewer #1 for pointing them out. We had shortened them and made the corrections accordingly. 

2. We thank Reviewer #1 for the suggestion to focus only on healthy older adults (Lines 391-395).

3. We thank Reviewer #1 for pointing them out. We had deleted the results of LF .

4. we had removed Section 5.2 part paragraph and focused on comparing exercise intensity resistance training.

5. We thank Reviewer #1 for the question. The manuscript had been removed and edited to conclusion. (line 502-509)

6. We thank Reviewer #1 for the comment. Conclusions are drawn from our findings and try to explain what kind of resistance training (intensity, load, type) is enough to elicit improvement in cardiovascular and autonomic health. (line 502-509)

We would like to thank the editor and reviewers for the time to review our manuscript. We replied the questions and suggestion in the following.

No.

Reviewer comment

Author response

Reviewer #1

The topic is very interesting and important because of the aging society. The study is interesting, and the results are promising. However, the main shortcoming of this paper is that it is not precise enough and confusing for the reader. The authors are switching between the topics. The main research question might be if low or high intensity resistance training elicit improvements in cardiovascular health. However, too many other aspects were raised and the authors do not adequately answer the main question. As the study is interesting, I recommend major revisions before considered for publication.

We appreciate that Reviewer #1 recognized the importance of aging and if low or high intensity resistance training elicit improvements in cardiovascular health. We will respond to all reasonable comments and the main question in the following.

1. General remark

The order of the references are not correct. I would recommend to substitute “subjects” with “participants” as this is more common today. The authors switch between cardiac autonomic activity, tone, control, and modulation. There should be an uniform term based on current literature. The discussion should be shortened. One page less would be still enough.

We appreciate the recognition from the Reviewer #1.The abstract section will not be modified for this time.

2. Introduction

Line 41+42: there should be a reference after the sentence “It is widely believed …”.

In general, the introduction is rather extensive, and the studies are described with too many details. The authors sometimes jump between the topics. For the reader who is unfamiliar with the ANS, it is difficult to follow. I would recommend being more precise and shorten the introduction. For example, it is not necessary to describe all studies in such detail.

We thank Reviewer #1 for the comment. As we responded to Comment #1, we carefully reviewed the Introduction multiple times and made significant revisions, and shorten the introduction. We had tried to precise enough and logical structure of the introduction. We hope that is satisfactory. ( Line 33-66, red words) Line 33-66

3. Material and methods

1.The authors should state how and from where the participants were recruited?

2.Were depression or adiposity an exclusion criteria?

3.How long was the resting time between the sets?

4.I miss some information concerning the autonomic measurements:

5.Time of day of the measurements

6.Sampling rate

Sampling rate

In my opinion it is not sufficient to write “… automatic and manual exclusion of artifact and ectopic beats based on recommendations of the Task Force”. The criteria for artifacts and ectopic beats should be clearly stated (e.g., specific length of RR interval).

Artifact cleaning method?

Were participants with specific number of artifacts excluded?

Why did the authors use 7.LF/HF as marker of sympatho-vagal balance although current literature does not support this assumption anymore (e.g., Billman, 2013, doi: 10.3389/fphys.2013.00026)?

We thank Reviewer #1 for the reminds. We will respond to all reasonable comments and the main question in the following.

1. Sixty non-smoking middle-aged and older adults (aged 52-68 years) (40 females, 20 males) were recruited from community. (Line 139)

2. The depression and adiposity were the exclusion criteria.(Line 143)

3. Subjects performed one set of 15 repetitions for each exercise to fatigue for the first four weeks of training, and two sets of 8 (HEX group) or 13 (LEX group) repetitions (with 2–3 min of rest between sets) thereafter at the proper resistance load. (Line 202)

4. Can be more explained and highlighted

5. in the morning, (Line 217-219)

6. We thank Reviewer #1 for pointing them out. We had made the corrections accordingly for artifacts and ectopic beats should be clearly stated (e.g., specific length of RR interval) (Line 235-238,  248-251). Seven subjects were excluded because of chronic inflammatory diseases (coronary artery disease, asthma, rheumatoid arthritis, poorly-controlled diabetes, and chronic renal disease) (Figure 1.)

7. The LF/HF ratio reflects the global sympatho-vagal balance, and serves as a measure of this balance. Whether or not this theory is still valid is still a matter of opinion.

There are also continuously related literatures to support the adoption. After consideration, we still follow the original experimental design for analysis.

Shaffer F, Ginsberg JP (2017) An overview of heart rate variability metrics and norms. Front public Heal 5:258. https://doi.org/10.3389/fpubh.2017.00258

PL Latchman, G Gates, J Pereira, R Axtell R(2020) The association between sympatho-vagal balance and central blood pressures. Physiol Int. 107(1):155-165. doi: 10.1556/2060.2020.00005

4.

Results

Page 8, line 298: “figure not shown” is not necessary or the authors include a figure.

Figure 2 is confusing. The authors should look for a diagram which is more intuitive for the reader.

1.We thank Reviewer #1 for pointing them out. We had added the title for figure1.

.

2. We thank Reviewer #1 for pointing this out.  Figure 2 can more intuitively show the Changes in the supine-standing ratio (SSR) as percentage differences between cardiovascular parameters and HRV parameters in pre-and post-study measurements for the three groups. Therefore we decided to keep the same results.

5.

Discussion

1.As already noted above, the discussion is too long. The authors should more focus on the assessed parameters in this study. E.g., VO2 max was not assessed and the sentences about this parameter could be deleted. Some sentences could also be deleted as they are only a repetition of the results (e.g., page 10, lines 356-359).

2.Moreover, as only relatively healthy older adults were measured, discussion should focus only on this group.

3.Section 5.2. should describe “vagal tone”, so why were the results of LF discussed? LF is no parameter for the vagal tone.

4.Section 5.3. is very long and confusing. The authors should be more precise. Some parts can be removed because they are basic physiological knowledge. Some parts are not relevant for this study.

5.The sense of the last paragraph (line 502-509) is not clear. Is it a summary of the study or of the literature (although no references were included)?

Line 522-533: this is the conclusion. It should be under a separate section.

6.As the main aim of the study was to compare low and high intensity resistance training, I am wondering why this aspect was so little discussed. It is clear that any kind of training elicits effects in untrained older adults. However, one important research question is, what kind of training (intensity, load, type) is adequate to elicit improvement in cardiovascular and autonomic health. In my opinion, the reader is interested in this question and therefore, the authors should focus on this in the discussion.

1.We thank Reviewer #1 for pointing them out. We had shortened them and made the corrections accordingly. 

2. We thank Reviewer #1 for the suggestion to focus only on healthy older adults (Lines 391-395).

3. We thank Reviewer #1 for pointing them out. We had deleted the results of LF .

4. we had removed Section 5.2 part paragraph and focused on comparing exercise intensity resistance training.

5. We thank Reviewer #1 for the question. The manuscript had been removed and edited to conclusion. (line 502-509)

6. We thank Reviewer #1 for the comment. Conclusions are drawn from our findings and try to explain what kind of resistance training (intensity, load, type) is enough to elicit improvement in cardiovascular and autonomic health. (line 502-509)

We would like to thank the editor and reviewers for the time to review our manuscript. We replied the questions and suggestion in the following.

No.

Reviewer comment

Author response

Reviewer #1

The topic is very interesting and important because of the aging society. The study is interesting, and the results are promising. However, the main shortcoming of this paper is that it is not precise enough and confusing for the reader. The authors are switching between the topics. The main research question might be if low or high intensity resistance training elicit improvements in cardiovascular health. However, too many other aspects were raised and the authors do not adequately answer the main question. As the study is interesting, I recommend major revisions before considered for publication.

We appreciate that Reviewer #1 recognized the importance of aging and if low or high intensity resistance training elicit improvements in cardiovascular health. We will respond to all reasonable comments and the main question in the following.

1. General remark

The order of the references are not correct. I would recommend to substitute “subjects” with “participants” as this is more common today. The authors switch between cardiac autonomic activity, tone, control, and modulation. There should be an uniform term based on current literature. The discussion should be shortened. One page less would be still enough.

We appreciate the recognition from the Reviewer #1.The abstract section will not be modified for this time.

2. Introduction

Line 41+42: there should be a reference after the sentence “It is widely believed …”.

In general, the introduction is rather extensive, and the studies are described with too many details. The authors sometimes jump between the topics. For the reader who is unfamiliar with the ANS, it is difficult to follow. I would recommend being more precise and shorten the introduction. For example, it is not necessary to describe all studies in such detail.

We thank Reviewer #1 for the comment. As we responded to Comment #1, we carefully reviewed the Introduction multiple times and made significant revisions, and shorten the introduction. We had tried to precise enough and logical structure of the introduction. We hope that is satisfactory. ( Line 33-66, red words) Line 33-66

3. Material and methods

1.The authors should state how and from where the participants were recruited?

2.Were depression or adiposity an exclusion criteria?

3.How long was the resting time between the sets?

4.I miss some information concerning the autonomic measurements:

5.Time of day of the measurements

6.Sampling rate

Sampling rate

In my opinion it is not sufficient to write “… automatic and manual exclusion of artifact and ectopic beats based on recommendations of the Task Force”. The criteria for artifacts and ectopic beats should be clearly stated (e.g., specific length of RR interval).

Artifact cleaning method?

Were participants with specific number of artifacts excluded?

Why did the authors use 7.LF/HF as marker of sympatho-vagal balance although current literature does not support this assumption anymore (e.g., Billman, 2013, doi: 10.3389/fphys.2013.00026)?

We thank Reviewer #1 for the reminds. We will respond to all reasonable comments and the main question in the following.

1. Sixty non-smoking middle-aged and older adults (aged 52-68 years) (40 females, 20 males) were recruited from community. (Line 139)

2. The depression and adiposity were the exclusion criteria.(Line 143)

3. Subjects performed one set of 15 repetitions for each exercise to fatigue for the first four weeks of training, and two sets of 8 (HEX group) or 13 (LEX group) repetitions (with 2–3 min of rest between sets) thereafter at the proper resistance load. (Line 202)

4. Can be more explained and highlighted

5. in the morning, (Line 217-219)

6. We thank Reviewer #1 for pointing them out. We had made the corrections accordingly for artifacts and ectopic beats should be clearly stated (e.g., specific length of RR interval) (Line 235-238,  248-251). Seven subjects were excluded because of chronic inflammatory diseases (coronary artery disease, asthma, rheumatoid arthritis, poorly-controlled diabetes, and chronic renal disease) (Figure 1.)

7. The LF/HF ratio reflects the global sympatho-vagal balance, and serves as a measure of this balance. Whether or not this theory is still valid is still a matter of opinion.

There are also continuously related literatures to support the adoption. After consideration, we still follow the original experimental design for analysis.

Shaffer F, Ginsberg JP (2017) An overview of heart rate variability metrics and norms. Front public Heal 5:258. https://doi.org/10.3389/fpubh.2017.00258

PL Latchman, G Gates, J Pereira, R Axtell R(2020) The association between sympatho-vagal balance and central blood pressures. Physiol Int. 107(1):155-165. doi: 10.1556/2060.2020.00005

4.

Results

Page 8, line 298: “figure not shown” is not necessary or the authors include a figure.

Figure 2 is confusing. The authors should look for a diagram which is more intuitive for the reader.

1.We thank Reviewer #1 for pointing them out. We had added the title for figure1.

.

2. We thank Reviewer #1 for pointing this out.  Figure 2 can more intuitively show the Changes in the supine-standing ratio (SSR) as percentage differences between cardiovascular parameters and HRV parameters in pre-and post-study measurements for the three groups. Therefore we decided to keep the same results.

5.

Discussion

1.As already noted above, the discussion is too long. The authors should more focus on the assessed parameters in this study. E.g., VO2 max was not assessed and the sentences about this parameter could be deleted. Some sentences could also be deleted as they are only a repetition of the results (e.g., page 10, lines 356-359).

2.Moreover, as only relatively healthy older adults were measured, discussion should focus only on this group.

3.Section 5.2. should describe “vagal tone”, so why were the results of LF discussed? LF is no parameter for the vagal tone.

4.Section 5.3. is very long and confusing. The authors should be more precise. Some parts can be removed because they are basic physiological knowledge. Some parts are not relevant for this study.

5.The sense of the last paragraph (line 502-509) is not clear. Is it a summary of the study or of the literature (although no references were included)?

Line 522-533: this is the conclusion. It should be under a separate section.

6.As the main aim of the study was to compare low and high intensity resistance training, I am wondering why this aspect was so little discussed. It is clear that any kind of training elicits effects in untrained older adults. However, one important research question is, what kind of training (intensity, load, type) is adequate to elicit improvement in cardiovascular and autonomic health. In my opinion, the reader is interested in this question and therefore, the authors should focus on this in the discussion.

1.We thank Reviewer #1 for pointing them out. We had shortened them and made the corrections accordingly. 

2. We thank Reviewer #1 for the suggestion to focus only on healthy older adults (Lines 391-395).

3. We thank Reviewer #1 for pointing them out. We had deleted the results of LF .

4. we had removed Section 5.2 part paragraph and focused on comparing exercise intensity resistance training.

5. We thank Reviewer #1 for the question. The manuscript had been removed and edited to conclusion. (line 502-509)

6. We thank Reviewer #1 for the comment. Conclusions are drawn from our findings and try to explain what kind of resistance training (intensity, load, type) is enough to elicit improvement in cardiovascular and autonomic health. (line 502-509)

We would like to thank the editor and reviewers for the time to review our manuscript. We replied the questions and suggestion in the following.

No.

Reviewer comment

Author response

Reviewer #1

The topic is very interesting and important because of the aging society. The study is interesting, and the results are promising. However, the main shortcoming of this paper is that it is not precise enough and confusing for the reader. The authors are switching between the topics. The main research question might be if low or high intensity resistance training elicit improvements in cardiovascular health. However, too many other aspects were raised and the authors do not adequately answer the main question. As the study is interesting, I recommend major revisions before considered for publication.

We appreciate that Reviewer #1 recognized the importance of aging and if low or high intensity resistance training elicit improvements in cardiovascular health. We will respond to all reasonable comments and the main question in the following.

1. General remark

The order of the references are not correct. I would recommend to substitute “subjects” with “participants” as this is more common today. The authors switch between cardiac autonomic activity, tone, control, and modulation. There should be an uniform term based on current literature. The discussion should be shortened. One page less would be still enough.

We appreciate the recognition from the Reviewer #1.The abstract section will not be modified for this time.

2. Introduction

Line 41+42: there should be a reference after the sentence “It is widely believed …”.

In general, the introduction is rather extensive, and the studies are described with too many details. The authors sometimes jump between the topics. For the reader who is unfamiliar with the ANS, it is difficult to follow. I would recommend being more precise and shorten the introduction. For example, it is not necessary to describe all studies in such detail.

We thank Reviewer #1 for the comment. As we responded to Comment #1, we carefully reviewed the Introduction multiple times and made significant revisions, and shorten the introduction. We had tried to precise enough and logical structure of the introduction. We hope that is satisfactory. ( Line 33-66, red words) Line 33-66

3. Material and methods

1.The authors should state how and from where the participants were recruited?

2.Were depression or adiposity an exclusion criteria?

3.How long was the resting time between the sets?

4.I miss some information concerning the autonomic measurements:

5.Time of day of the measurements

6.Sampling rate

Sampling rate

In my opinion it is not sufficient to write “… automatic and manual exclusion of artifact and ectopic beats based on recommendations of the Task Force”. The criteria for artifacts and ectopic beats should be clearly stated (e.g., specific length of RR interval).

Artifact cleaning method?

Were participants with specific number of artifacts excluded?

Why did the authors use 7.LF/HF as marker of sympatho-vagal balance although current literature does not support this assumption anymore (e.g., Billman, 2013, doi: 10.3389/fphys.2013.00026)?

We thank Reviewer #1 for the reminds. We will respond to all reasonable comments and the main question in the following.

1. Sixty non-smoking middle-aged and older adults (aged 52-68 years) (40 females, 20 males) were recruited from community. (Line 139)

2. The depression and adiposity were the exclusion criteria.(Line 143)

3. Subjects performed one set of 15 repetitions for each exercise to fatigue for the first four weeks of training, and two sets of 8 (HEX group) or 13 (LEX group) repetitions (with 2–3 min of rest between sets) thereafter at the proper resistance load. (Line 202)

4. Can be more explained and highlighted

5. in the morning, (Line 217-219)

6. We thank Reviewer #1 for pointing them out. We had made the corrections accordingly for artifacts and ectopic beats should be clearly stated (e.g., specific length of RR interval) (Line 235-238,  248-251). Seven subjects were excluded because of chronic inflammatory diseases (coronary artery disease, asthma, rheumatoid arthritis, poorly-controlled diabetes, and chronic renal disease) (Figure 1.)

7. The LF/HF ratio reflects the global sympatho-vagal balance, and serves as a measure of this balance. Whether or not this theory is still valid is still a matter of opinion.

There are also continuously related literatures to support the adoption. After consideration, we still follow the original experimental design for analysis.

Shaffer F, Ginsberg JP (2017) An overview of heart rate variability metrics and norms. Front public Heal 5:258. https://doi.org/10.3389/fpubh.2017.00258

PL Latchman, G Gates, J Pereira, R Axtell R(2020) The association between sympatho-vagal balance and central blood pressures. Physiol Int. 107(1):155-165. doi: 10.1556/2060.2020.00005

4.

Results

Page 8, line 298: “figure not shown” is not necessary or the authors include a figure.

Figure 2 is confusing. The authors should look for a diagram which is more intuitive for the reader.

1.We thank Reviewer #1 for pointing them out. We had added the title for figure1.

.

2. We thank Reviewer #1 for pointing this out.  Figure 2 can more intuitively show the Changes in the supine-standing ratio (SSR) as percentage differences between cardiovascular parameters and HRV parameters in pre-and post-study measurements for the three groups. Therefore we decided to keep the same results.

5.

Discussion

1.As already noted above, the discussion is too long. The authors should more focus on the assessed parameters in this study. E.g., VO2 max was not assessed and the sentences about this parameter could be deleted. Some sentences could also be deleted as they are only a repetition of the results (e.g., page 10, lines 356-359).

2.Moreover, as only relatively healthy older adults were measured, discussion should focus only on this group.

3.Section 5.2. should describe “vagal tone”, so why were the results of LF discussed? LF is no parameter for the vagal tone.

4.Section 5.3. is very long and confusing. The authors should be more precise. Some parts can be removed because they are basic physiological knowledge. Some parts are not relevant for this study.

5.The sense of the last paragraph (line 502-509) is not clear. Is it a summary of the study or of the literature (although no references were included)?

Line 522-533: this is the conclusion. It should be under a separate section.

6.As the main aim of the study was to compare low and high intensity resistance training, I am wondering why this aspect was so little discussed. It is clear that any kind of training elicits effects in untrained older adults. However, one important research question is, what kind of training (intensity, load, type) is adequate to elicit improvement in cardiovascular and autonomic health. In my opinion, the reader is interested in this question and therefore, the authors should focus on this in the discussion.

1.We thank Reviewer #1 for pointing them out. We had shortened them and made the corrections accordingly. 

2. We thank Reviewer #1 for the suggestion to focus only on healthy older adults (Lines 391-395).

3. We thank Reviewer #1 for pointing them out. We had deleted the results of LF .

4. we had removed Section 5.2 part paragraph and focused on comparing exercise intensity resistance training.

5. We thank Reviewer #1 for the question. The manuscript had been removed and edited to conclusion. (line 502-509)

6. We thank Reviewer #1 for the comment. Conclusions are drawn from our findings and try to explain what kind of resistance training (intensity, load, type) is enough to elicit improvement in cardiovascular and autonomic health. (line 502-509)

We would like to thank the editor and reviewers for the time to review our manuscript. We replied the questions and suggestion in the following.

No.

Reviewer comment

Author response

Reviewer #1

The topic is very interesting and important because of the aging society. The study is interesting, and the results are promising. However, the main shortcoming of this paper is that it is not precise enough and confusing for the reader. The authors are switching between the topics. The main research question might be if low or high intensity resistance training elicit improvements in cardiovascular health. However, too many other aspects were raised and the authors do not adequately answer the main question. As the study is interesting, I recommend major revisions before considered for publication.

We appreciate that Reviewer #1 recognized the importance of aging and if low or high intensity resistance training elicit improvements in cardiovascular health. We will respond to all reasonable comments and the main question in the following.

1. General remark

The order of the references are not correct. I would recommend to substitute “subjects” with “participants” as this is more common today. The authors switch between cardiac autonomic activity, tone, control, and modulation. There should be an uniform term based on current literature. The discussion should be shortened. One page less would be still enough.

We appreciate the recognition from the Reviewer #1.The abstract section will not be modified for this time.

2. Introduction

Line 41+42: there should be a reference after the sentence “It is widely believed …”.

In general, the introduction is rather extensive, and the studies are described with too many details. The authors sometimes jump between the topics. For the reader who is unfamiliar with the ANS, it is difficult to follow. I would recommend being more precise and shorten the introduction. For example, it is not necessary to describe all studies in such detail.

We thank Reviewer #1 for the comment. As we responded to Comment #1, we carefully reviewed the Introduction multiple times and made significant revisions, and shorten the introduction. We had tried to precise enough and logical structure of the introduction. We hope that is satisfactory. ( Line 33-66, red words) Line 33-66

3. Material and methods

1.The authors should state how and from where the participants were recruited?

2.Were depression or adiposity an exclusion criteria?

3.How long was the resting time between the sets?

4.I miss some information concerning the autonomic measurements:

5.Time of day of the measurements

6.Sampling rate

Sampling rate

In my opinion it is not sufficient to write “… automatic and manual exclusion of artifact and ectopic beats based on recommendations of the Task Force”. The criteria for artifacts and ectopic beats should be clearly stated (e.g., specific length of RR interval).

Artifact cleaning method?

Were participants with specific number of artifacts excluded?

Why did the authors use 7.LF/HF as marker of sympatho-vagal balance although current literature does not support this assumption anymore (e.g., Billman, 2013, doi: 10.3389/fphys.2013.00026)?

We thank Reviewer #1 for the reminds. We will respond to all reasonable comments and the main question in the following.

1. Sixty non-smoking middle-aged and older adults (aged 52-68 years) (40 females, 20 males) were recruited from community. (Line 139)

2. The depression and adiposity were the exclusion criteria.(Line 143)

3. Subjects performed one set of 15 repetitions for each exercise to fatigue for the first four weeks of training, and two sets of 8 (HEX group) or 13 (LEX group) repetitions (with 2–3 min of rest between sets) thereafter at the proper resistance load. (Line 202)

4. Can be more explained and highlighted

5. in the morning, (Line 217-219)

6. We thank Reviewer #1 for pointing them out. We had made the corrections accordingly for artifacts and ectopic beats should be clearly stated (e.g., specific length of RR interval) (Line 235-238,  248-251). Seven subjects were excluded because of chronic inflammatory diseases (coronary artery disease, asthma, rheumatoid arthritis, poorly-controlled diabetes, and chronic renal disease) (Figure 1.)

7. The LF/HF ratio reflects the global sympatho-vagal balance, and serves as a measure of this balance. Whether or not this theory is still valid is still a matter of opinion.

There are also continuously related literatures to support the adoption. After consideration, we still follow the original experimental design for analysis.

Shaffer F, Ginsberg JP (2017) An overview of heart rate variability metrics and norms. Front public Heal 5:258. https://doi.org/10.3389/fpubh.2017.00258

PL Latchman, G Gates, J Pereira, R Axtell R(2020) The association between sympatho-vagal balance and central blood pressures. Physiol Int. 107(1):155-165. doi: 10.1556/2060.2020.00005

4.

Results

Page 8, line 298: “figure not shown” is not necessary or the authors include a figure.

Figure 2 is confusing. The authors should look for a diagram which is more intuitive for the reader.

1.We thank Reviewer #1 for pointing them out. We had added the title for figure1.

.

2. We thank Reviewer #1 for pointing this out.  Figure 2 can more intuitively show the Changes in the supine-standing ratio (SSR) as percentage differences between cardiovascular parameters and HRV parameters in pre-and post-study measurements for the three groups. Therefore we decided to keep the same results.

5.

Discussion

1.As already noted above, the discussion is too long. The authors should more focus on the assessed parameters in this study. E.g., VO2 max was not assessed and the sentences about this parameter could be deleted. Some sentences could also be deleted as they are only a repetition of the results (e.g., page 10, lines 356-359).

2.Moreover, as only relatively healthy older adults were measured, discussion should focus only on this group.

3.Section 5.2. should describe “vagal tone”, so why were the results of LF discussed? LF is no parameter for the vagal tone.

4.Section 5.3. is very long and confusing. The authors should be more precise. Some parts can be removed because they are basic physiological knowledge. Some parts are not relevant for this study.

5.The sense of the last paragraph (line 502-509) is not clear. Is it a summary of the study or of the literature (although no references were included)?

Line 522-533: this is the conclusion. It should be under a separate section.

6.As the main aim of the study was to compare low and high intensity resistance training, I am wondering why this aspect was so little discussed. It is clear that any kind of training elicits effects in untrained older adults. However, one important research question is, what kind of training (intensity, load, type) is adequate to elicit improvement in cardiovascular and autonomic health. In my opinion, the reader is interested in this question and therefore, the authors should focus on this in the discussion.

1.We thank Reviewer #1 for pointing them out. We had shortened them and made the corrections accordingly. 

2. We thank Reviewer #1 for the suggestion to focus only on healthy older adults (Lines 391-395).

3. We thank Reviewer #1 for pointing them out. We had deleted the results of LF .

4. we had removed Section 5.2 part paragraph and focused on comparing exercise intensity resistance training.

5. We thank Reviewer #1 for the question. The manuscript had been removed and edited to conclusion. (line 502-509)

6. We thank Reviewer #1 for the comment. Conclusions are drawn from our findings and try to explain what kind of resistance training (intensity, load, type) is enough to elicit improvement in cardiovascular and autonomic health. (line 502-509)

Reviewer 2 Report

This trial aimed to to evaluate the effects of 24 weeks of resistance exercise training of various intensities on muscular strength, hemodynamics, and HRV at rest and in response to orthostatic maneuvers in middle-aged and older adults and was hypothesized that high intensity resistance training would be more effective than low-moderate intensity resistance training.

There are several major concerns related to this manuscript and therefore I am afraid that it did not merit a high enough rating in order to be published in International Journal. Environmental Research and Public Health.

#1: Believe that in the title, the fact that the participants have arterial hypertension and diabetes mellitus should be included, since they are not healthy.

#2: Were there previous guidelines on avoiding physical activity, not injecting foods that contain caffeine and controlling the environment for HRV collection? I believe that this is important information that should be included in the methods secction.

#3: Was any sample calculation performed for this protocol?

#4: Where were these individuals screened from? This was not clear in the methods.

#5: Item 2.2 entitled Study design, there is no description of the type of study, please include it.

#6: Were there any adverse events during the development of the study? Or some variable evaluated in relation to the effort exerted by these individuals? (e.g BORG scale)

#7: Absence of the name of the figure 1. In addition, I indicate that the description of the captions and analyzes in figure 2, be placed after the title of figure 2.

#8: According to the changes found in the cardiovascular analyses, training was shown to be beneficial for the cardiovascular system, but was there any evaluation beyond HRV under this system?

#9: We know that functional capacity can be evaluated according to several available tools, and that when it comes to the evaluation of peripheral muscle strength, it is ideal that it be evaluated by the gold standard (isokinetic dynamometer), considering that the evaluation was used by the 1RM, I believe that this should be assumed as a limitation of the study and discussed as well.

#10: Before describing the limitations of the study, I believe that there should be a paragraph, showing that high intensity resistance training would be more effective than low-moderate intensity resistance training, since this is the purpose of the study.

Author Response

We would like to thank the editor and reviewers for the time to review our manuscript. We replied the questions and suggestion in the following.

No.

Reviewer comment

Author response

Reviewer #2

1.

Believe that in the title, the fact that the participants have arterial hypertension and diabetes mellitus should be included, since they are not healthy.

We thank Reviewer #2 for the question. We recruit healthy elderly people without serious diseases from the community to ensure that the subjects were free from any cardiovascular or orthopedic/neuromuscular diseases which might hinder them from doing resistance training. (LINE 138-143). There are usually some chronic diseases in middle-aged and elderly adults, but they are not this study variable or key factors affecting the research. It should not be included title. Please understand our thought.

2.

Were there previous guidelines on avoiding physical activity, not injecting foods that contain caffeine and controlling the environment for HRV collection? I believe that this is important information that should be included in the methods secction.

We thank Reviewer #2 for the suggestion. We adjust the paragraph of the section 2.1 participants to provide greater details about the guidelines (L158-160). Please refer to the text again.

3.

Was any sample calculation performed for this protocol?

We thank Reviewer #2 for the comment. Yes, we had calculated the sample size at least 35 subjects via G-power for this protocol.(L283-285)

4.

Where were these individuals screened from? This was not clear in the methods.

We thank Reviewer #2 for the comment. We add the “who were recruited from community centers in southern Taiwan” (L139). Please refer to the text again.

5.

Item 2.2 entitled Study design, there is no description of the type of study, please include it.

We thank Reviewer #2 for the comment.  We adjust the paragraph of the section on 2.2 entitled study design. Please refer to the text again.( line 164-173)

6.

Were there any adverse events during the development of the study? Or some variable evaluated in relation to the effort exerted by these individuals? (e.g BORG scale)

We thank Reviewer #2 for the comment. There were six subjects were excluded because of chronic inflammatory diseases (coronary artery disease, asthma, rheumatoid arthritis, poorly-controlled diabetes, and chronic renal disease), and one female was excluded as she was already performing resistance training at the time. Please refer to the text again.( line 151-154). Monitoring exercise intensity during resistance training using the session RPE scale(line 213-214).

7.

Absence of the name of the figure 1. In addition, I indicate that the description of the captions and analyzes in figure 2, be placed after the title of figure 2.

We thank Reviewer #2 for the suggestion. We adjust the captions in figure 1 and 2(lin286,347-351).

8.

According to the changes found in the cardiovascular analyses, training was shown to be beneficial for the cardiovascular system, but was there any evaluation beyond HRV under this system?

We thank Reviewer #2 for pointing them out. We had add the description accordingly. (line503-509).

9.

We know that functional capacity can be evaluated according to several available tools, and that when it comes to the evaluation of peripheral muscle strength, it is ideal that it be evaluated by the gold standard (isokinetic dynamometer), considering that the evaluation was used by the 1RM, I believe that this should be assumed as a limitation of the study and discussed as well.

We thank Reviewer #2 for the suggestion. We did address the choice of testing as a limitation of the study.

Line(580-586).

10.

Before describing the limitations of the study, I believe that there should be a paragraph, showing that high intensity resistance training would be more effective than low-moderate intensity resistance training, since this is the purpose of the study.

We thank Reviewer #2 for the suggestion. We made some brief description before the limitation(lin563-566).